# Numerical Investigation on Unsteady Separation Flow Control in an Axial Compressor Using Detached-Eddy Simulation

**Mingming Zhang [1] and Anping Hou [2,\*]**

[1]  College of Applied Sciences, Beijing University of Technology, Beijing 100124, China
[2]  School of Energy and Power, Beihang University, Beijing 100191, China
\*  Correspondence: houap@buaa.edu.cn; Tel.: +86-010-8231-6624

**Abstract:** Unsteady excitation has proved its effectiveness in separation flow control and has been extensively studied. It is observed that disordered shedding vortices in compressors can be controlled by unsteady excitation, especially when the excitation frequency coincides with the frequency of the shedding vortex. Furthermore, former experimental results indicated that unsteady excitation at other frequencies also had an impact on the structure of shedding vortices. To investigate the impact of excitation frequency on vortex shedding structure, the Detached-Eddy Simulation (DES) method was applied in the simulation of shedding vortex structure under unsteady excitations at different frequencies in an axial compressor. Effectiveness of the DES method was proved by comparison with URANS results. The simulation results showed a good agreement with the former experiment. The numerical results indicated that the separation flow can be partly controlled when the excitation frequency coincided with the unsteady flow inherent frequency. It showed an increase in stage performance under the less-studied separation flow control by excitation at a certain frequency of pressure side shedding vortex. Compared with other frequencies of shedding vortices, the frequency of pressure side shedding vortex was less sensitive to mass-flow variation. Therefore, it has potential for easier application on flow control in industrial compressors.

**Keywords:** unsteady flow control; vortex dynamics; separation flow; Detached-Eddy Simulation

## 1. Introduction

The flow separation at the compressor trailing edge is unavoidable due to the high adverse pressure gradient, especially in modern gas turbines which have an increased loading than in the past. Investigations in turbomachinery have shown that the separation vortex is one of the main sources of loss at near stall point, and it can reduce the stage efficiency as well as the stall range. Therefore, methods aiming to control and decrease the separation have been extensively studied.

Analysis of separation flow started in the 1960s with experiments in airfoils with a large attack angle [1]. In the primary stage, passive control methods were analyzed, and then the active controls of separation flow were considered. Active control methods such as suction blowing on the blade surface were confirmed to have a better capacity than passive control [2]. Further development of the active control method was unsteady excitation control. Dynamic excitation control methods can provide more performance improvement with less injected mass flow than the constant excitation control [3–6], because the separation flow structure at large attack angles is inherently unsteady. To be precise, the structure of vortex shedding is physically similar to the Karman vortex street structure [7,8].

Under different types of unsteady excitation control methods, the separated flow can be controlled to be in order, including unsteady suction blowing, upstream wake excitation, oscillating guide



vanes, and total pressure fluctuation in the incoming flow by trumpets [9–11]. In an axial compressor stage, it had been increased to a maximum of 40% of the stall margin by dynamic air injectors in the experiment. The mass-flow rate in these air injectors was less than 1% of the compressor flow rate [10]. Koc applied plasma actuators to provide an active separation control on a bluff body, and a "locked-on" effect was shown when the excitation frequency approached the natural vortex shedding frequency [11]. Experimental [12] and numerical analysis [13] in an axial compressor test rig demonstrated that the stall boundary, pressure rise, and near-stall efficiency were all increased with unsteady excitation by an appropriate frequency. The maximum enhancements under separation control were 5.4% increasement in total pressure rise, 5.5% increasement in the efficiency of the compressor, and 30.7% increasement in the relative stall margin. In these studies, the maximum enhancement on stage performance as well as stall range could be acquired when the dynamic excitation frequency was in coincidence with the vortex shedding frequency.

However, there are few investigations that focus on the mechanism of non-vortex-shedding frequency excitation, which was also proved to have a remarkable effect on cascade performance [12]. For the sake of explanation on the experimental phenomenon and improvement of the current unsteady separation control theory, the Detached-Eddy Simulation (DES) method was applied in this paper as a high-fidelity numerical simulation tool. The same low-speed axial compressor test rig as used by Li [12] was applied. In order to analyze the impact of excitation on vortex shedding structure, this study analyzed the performances under unsteady excitations at different frequencies in the compressor model.

Numerical results that were confirmed by previous experimental works were composed of three classifications of separation flow control under unsteady excitations: Vortex-shedding control (VSC), suction-side separation vortex control (SSVC), and pressure-side separation vortex control (PSVC). It could be concluded that these separation control methods were effective when the excitation frequency coincided with the inherently unsteady flow frequency. In this test case, the enhancement of stage performance can be achieved at the whole working range by excitation at a certain frequency of pressure side shedding vortex. Compared with traditional unsteady separation control, the new unsteady excitation method may be easy to apply in industrial compressors and has a great potential.

## 2. Numerical Approach

The DES method based on the SST k-ω turbulence model [14–16] was used in this study. Detached-Eddy Simulation (DES) is a hybrid large eddy simulation (LES)/RANS method. The concept of DES is to calculate the boundary layer by a RANS turbulence model and to switch the turbulence model into a LES mode in detached regions. Compared with using LES in the global computational domain, the DES method not only ensures the calculation accuracy, but also improves the calculation efficiency and saves the calculation cost. Because of the balance of computation resources and fidelity, the DES method had been widely applied in capturing vortex-shedding structures in turbomachines [17–19]. In the DES calculation, the LES model is activated in the region where the turbulence length predicted by the RANS model is larger than the local mesh scale. The turbulence length scale in an SST k-ω model and in the DES model are shown in Equations (1) and (2), respectively:

$$l_{k-\omega} = k^{0.5}/(\beta \times \omega), \tag{1}$$

$$l_{DES} = \min(l_{k-\omega}, C_{DES}\Delta). \tag{2}$$

The dissipative term of the k-transport equation, the only modified term in the DES model, is transferred to Equation (3):

$$D_{DES}^{k} = \rho k^{2/3}/l_{DES}. \tag{3}$$

The commercial CFD software ANSYS CFX was used for the DES calculation. The advection term was discretized by the High Resolution Scheme, and the Second Order Backward Euler method was applied in the discretization of transient terms. According to the experimental results, the highest

frequency of interest in this unsteady phenomenon of this stage would be less than 2500 Hz, so there must be more than 40 time steps in a blade computing pitch. In the unsteady simulation, every blade passing pitch was divided into 105 time steps with each time step including a maximum of 50 inner interaction steps. In this paper, total performance parameters and separation positions were dealt with time-averaged results recorded after a stable quasi-periodic flow state achieved. Fast Fourier Transformation analysis was based on the data collected in over 200 blade-passing pitch cycles.

## 3. Compressor Test Model

The rotor stage of a low-speed axial compressor test rig shown in Figure 1 in Beihang University [12] was applied in this analysis. The test rig consists of a 13-blade stator row and a 19-blade rotor row with the C4 profile (Figure 2). At the mid-span, the rotor blade has a chord length of 52 mm, a solidity of 0.605, a stagger angle of 34.49 degree, and an outlet angle of 41.99 degree. The design speed of the test rig is 3000 rpm, and the mass flow is 2.40 m³/s at the design point with a total pressure rise equal to 1500 Pa. A detailed introduction of the test rig is presented in Li [12] and Zhang [20]. In this study, a blade cascade extended from the rotor's mid-span profile was applied as the compressor test model. The length in the vertical direction was set as 20% span according to the literature.

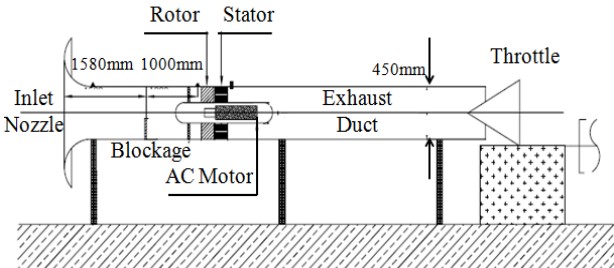

**Figure 1.** Compressor test rig.

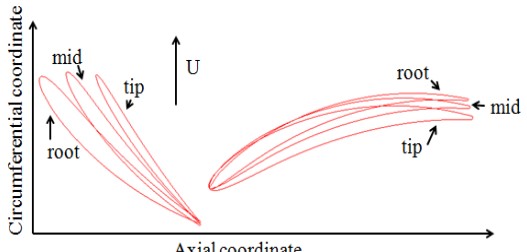

**Figure 2.** Blade sections at three spanwise locations.

An axial total pressure fluctuation in the inlet boundary was added as the source of unsteady excitation, which was set to physically imitate the periodic sound excitation generated by pneumatic speakers. The schematic plot of the sound generator and the figure in the experiment by Li [12] are shown in Figure 3.

To imitate the inlet unsteady excitation, the total pressure at inlet condition was set as Equation (4), where $P_{inlet}$ was the total pressure on the inlet boundary, $t$ was the physical time during the unsteady simulation, $A_m$ was the maximum excitation amplitude of pressure, and $f$ was the frequency of the excitation. Total temperature and velocity at inlet directions were kept constant in the computations.

$$P_{inlet} = 101325 + A_m \times \sin(f \times 2\pi \times t). \tag{4}$$

The maximum excitation amplitude of pressure was set to be 600 Pa in the experiment, and the maximum flow control effect under unsteady excitation could be obtained at that amplitude. In the computations, the maximum amplitude was extended to 1000 Pa. The computation boundary conditions were set up based on the parameters of a near-stall working state with a total pressure

rise over 1600 Pa in the calculation. Periodic boundary conditions were set between the up and down side of the computation domain as well as between the left and right side. The total pressure boundary condition was used at the inlet and the averaged static pressure boundary condition was set to the outlet.

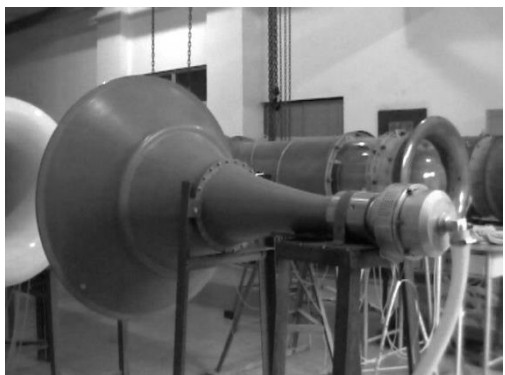
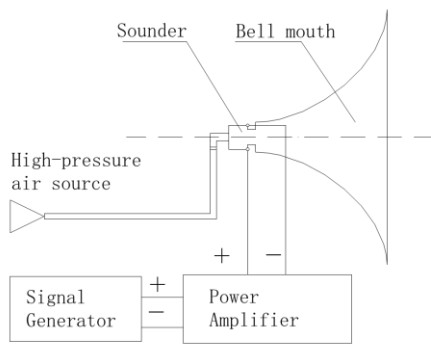

**Figure 3.** The schematic plot of the sound generator in the experiment.

Since DES results can be sensitive to grid resolution, four sets of mesh solutions with the same topology were studied to guarantee the accuracy of results. The Y-plus of the meshes near the blade surface were kept smaller than one in all four cases. The separation positions were chosen as the reference of shedding vortex structure (identification of separation position would be described in Section 4), and the results with the four meshes were shown in Figure 4. When the amount of mesh grids exceeded 7.36 million, the simulated separation positions were beginning to converge. Therefore, the mesh solution with 7.36 million mesh grids was chosen as the basic solution, which contained 347 nodes in the axial direction, 134 nodes in the pitchwise direction, and 125 nodes in the spanwise direction near the blade region.

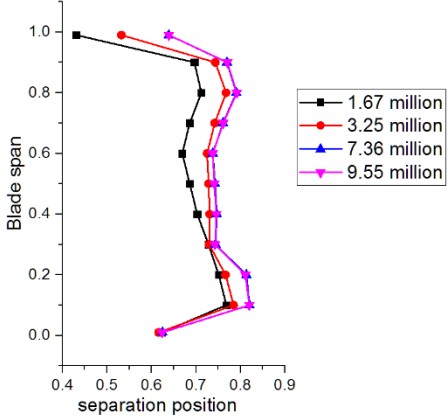

**Figure 4.** Separation positions with different grids.

From the numerical results, it could be concluded that the separation positions were varying less than 5% between 30% and 70% of the blade span, which indicated that the vortex shedding flow could be possibly considered as a quasi-3D phenomenon in circumferential direction at the middle span of this blade. In order to exclude the influence of hub/shroud secondary flow and focus on the separation flow, a 1 mm-thick curved slice at 50% span of the rotor blade was applied as a quasi-3D mode. Reliability of the quasi-3D model on the simulation of shedding vortex structure by high-fidelity simulation was verified by Zhao [21]. The quasi-3D model shared the same mesh grid distribution in the B2B section with the 3D mesh grid. The free slip boundary conditions were applied on the top and bottom surfaces of the quasi-3D model.

## 4. Results

### 4.1. Analysis on the Inherent Unsteady Flow Structure

The natural type of unsteady separation flow structure was analyzed as a reference state in this section. The DES simulations have been performed at different mass-flow coefficients. Characteristic lines of the quasi-3D model were shown in Figure 5. The mass-flow coefficient was defined as the ratio of averaged axial velocity to averaged total velocity. The pressure coefficient $\Delta Cp$ was defined as Equation (5):

$$\Delta Cp = \Delta p / \left(\frac{1}{2}\rho U_m^2\right) \tag{5}$$

where $\Delta p$ was pressure raise, $U_m$ was the rotational speed at midspan, and $\rho$ was the density.

In continuum mechanics, the vorticity $\vec{\omega}$ is a pseudovector field defined as the curl of the flow velocity $\vec{u}$ vector. It describes the local spinning motion of a continuum near some point. The definition can be expressed by the vector analysis formula:

$$\vec{\omega} \equiv \nabla \times \vec{u} \tag{6}$$

where $\nabla$ is the del operator. The vorticity of a two-dimensional flow is always perpendicular to the plane of the flow, and therefore can be considered a scalar field. In this paper the working point A near the stall was selected as the research working point to investigate unsteady separation flow structure. The instantaneous vorticity contour at the working point A was presented in Figure 6. As the graph showed, the separation on the suction side started near the middle length of chord, and then the separation vortex spread downstream and induced a shedding vortex at the trailing edge of blade.

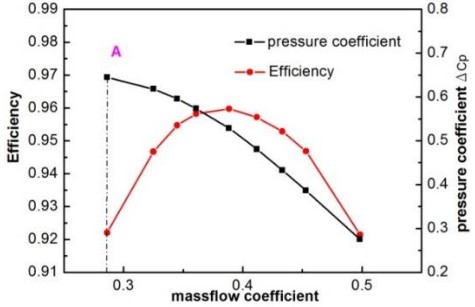

**Figure 5.** Characteristic lines of the quasi-3D model.

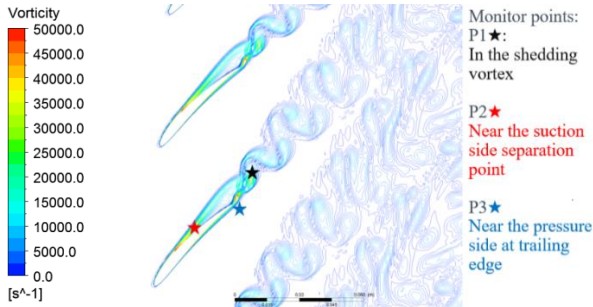

**Figure 6.** Instantaneous vorticity contour at working point A.

The vortex shedding phenomenon at the blade trailing edge, which was physically similar to the Karman vortex street, could be described as that separation vortices on the two sides of the blade combined with each other at the trailing edge with different vorticity directions, and then shed off in pairs when the balance of vorticity was obtained. Analyzed from this physical mechanism, three different types of unsteady flows existed in the flow field: Shedding vortex, suction-side separation

vortex, and pressure-side separation vortex. To obtain the characteristic frequencies of the three inherently unsteady flows, the analysis on vorticity fluctuations on monitor points P1, P2, and P3 (marked in Figure 6) was conducted. The vorticity fluctuations in the time domain and corresponding spectrum analysis at the three monitor points at working point A are shown in Figure 7.

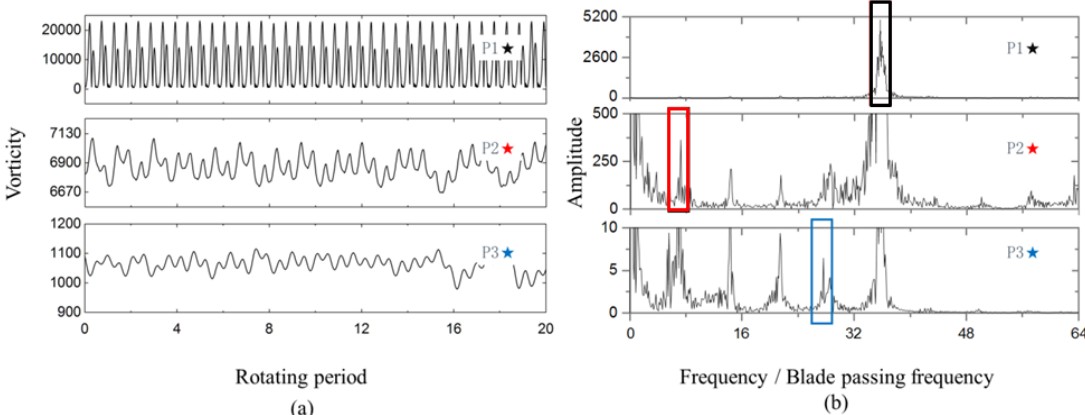

**Figure 7.** Vorticity fluctuations in time domain (**a**) and frequency domain (**b**) at monitor points.

A dominant frequency can be identified in all three locations, which was 36.5 times the rotation frequency (RF). The maximum amplitude at this frequency was located in the region of shedding vortices, which signified that it was the frequency of shedding vortex $F_{shed}$. The dominant frequency marked by the red square reached its maximum amplitude near the separation point on the suction side, which indicated the suction-side vortex separated at 7.12 times the rotation frequency ($F_{ss}$, frequency of suction-side vortex shedding). The dominant frequency at 27.6 times the rotation frequency (marked by the blue square) reached its maximum amplitude near the trailing edge in the pressure side. Based on the physical mechanism, this frequency was the inherent frequency of the pressure-side separation vortex named as $F_{ps}$.

Variations of the frequencies to mass-flow coefficients are shown in Figure 8 for the three types of unsteady separation flows. It could be concluded that the vortex shedding frequency $F_{shed}$ varied more than 100% with mass-flow coefficients, while the frequencies $F_{ps}$ and $F_{ss}$ were insensitive to the variation of working conditions.

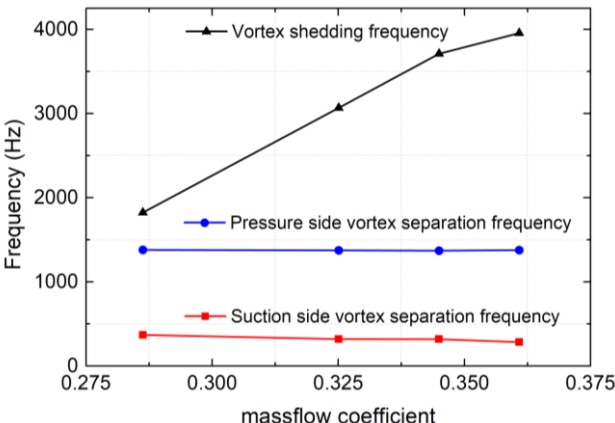

**Figure 8.** Variation of inherent frequencies with mass flow.

It had to be mentioned that the vortex shedding frequency measured by experiments [12] at the near stall point was 36.8 times the rotation frequency (36.8 RF). The agreement in $F_{shed}$ between numerical and experimental approaches indicated that the method used in this research was sure in capturing the vortex shedding structure of separation flow. Thus, it was chosen as working point A

to investigate separation flow characteristics in this paper. And the experimental results at the near stall point in Reference [12] are shown in the following section to verify the phenomenon shown in numerical studies.

### 4.2. Identification of Separation Position

Time-averaged wall shear stress distribution on the rotor blade surface at working point A is shown in Figure 9. The separation position on the suction side was identified where the wall shear stress on the suction side switched from positive to negative. At working point A, the flow on the suction side separated at 43.5% of the chord length from the leading edge. Figure 10 shows the variation of separation positions to the mass-flow coefficients. The *y*-axis label of the separation position was the chordwise of the blade. The vortices separated almost at the trailing edge of the blade in the large flow rate. The separation on the suction-side separation emerged distinctly when the mass-flow coefficient was less than 0.42. And the separation position moved towards the leading edge with the decrease of the mass-flow coefficient because of the increase in the blade attack angle.

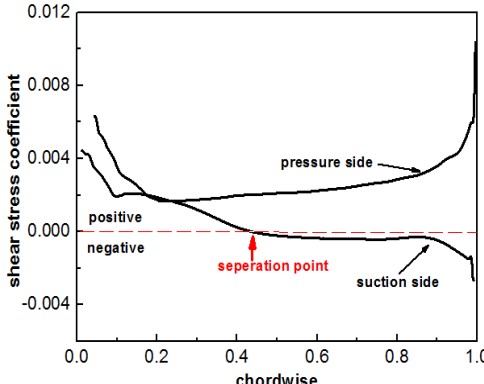

**Figure 9.** Wall shear stress distribution on the rotor blade surface.

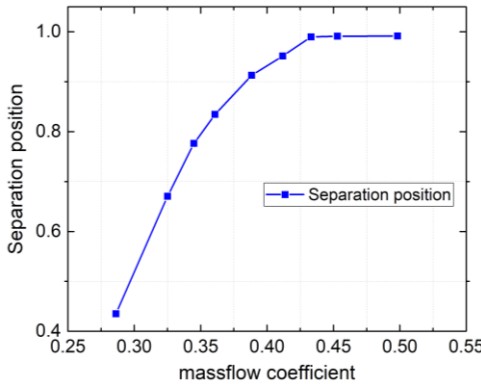

**Figure 10.** Variation of the separation position with mass flow.

To investigate the loss generated by separation flow quantitatively, 21 monitor points of total pressure were distributed uniformly in the circumferential direction at the middle span of rotor exit. With the decrease of the mass-flow coefficient, the area of total pressure loss was in expansion with the increase of the minimum amplitude of the pressure coefficient, as shown in Figure 11. The phenomenon was consistent with conclusions from previous studies that the loss generated by separation flow increased with the decrease of the mass-flow coefficient. In the next section, the changes of loss in distribution under unsteady excitations will be presented.

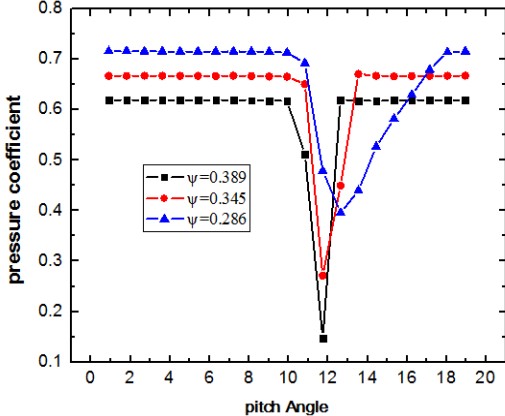

**Figure 11.** Pressure distribution at pitchwise to different mass-flow coefficients.

### 4.3. Selection of Parameters of Excitation

Unsteady excitations can cause an influence on mass-flow rate as well as pressure raise and efficiency, which makes it difficult to make a comparison between the effects under different excitation frequencies. To make a quantitative comparison, separation position on the suction side was chosen as the characteristic parameter for evaluating the effectiveness of separation control. Excitation, which had positive effects on stage performance, would translocate the separation position to the trailing edge. It meant that the separated flow was restructured, and delayed. Total pressure distributions of the blade downstream and vorticity contours were also presented to analyze the transformation of pressure loss regions. Investigations in this section were all carried at working point A, and all excitation frequencies were set as integral multiples of RF.

According to previous studies [9–13], excitation at the vortex shedding frequency had a remarkable effect on control of the flow field. Therefore, excitations with different amplitudes at the same frequency of 36 times the RF were first investigated to analyze the effect of excitation amplitude $A_m$ on the control of the flow field.

As indicated in Figure 12, the separation position kept moving to the trailing edge with the increase in excitation amplitude $A_m$. And the efficiency under different excitations were all increased. Based on the results, the positive influence on the separation control and efficiency enhancement can be obtained at excitations with computed amplitudes. But there existed a peak efficiency point that indicated the maximum benefit was gained for the performance by the excitation condition. In the following research, 400 Pa was chosen as the amplitude of unsteady excitation because of the maximum efficiency improvement in this condition.

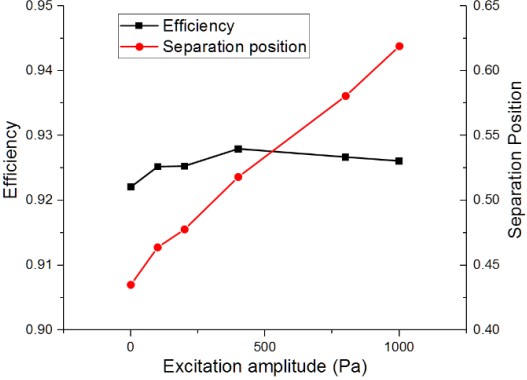

**Figure 12.** Influence of excitation amplitude on efficiency and separation position.

Then the frequency of excitation was varied from 0 to 50 times the RF, which contained 27 quantities. And the effects of these frequencies on the control of separation vortex structure are shown

in Figure 13. From the figure, it was concluded that the positive effects on the vortex shedding were obtained when the frequencies of excitation were closed to harmonic frequencies of $F_{shed}$ (marked by the black line) and harmonic frequencies of $F_{ps}$ (marked by the blue line). Excitations with the frequencies near $F_{ss}$ and its harmonic frequencies (marked by the red lines) could cause a dominant negative effect on the separation control. These phenomena can be verified by experimental results on the variation of relative efficiency at the near stall point [12]. For the difference of frequencies and effects between the experimental and computed results, there were two possible causes. Because of the limitation of CFD in simulating the turbulent flow, the predicted structure of vortices still had a little discrepancy in the experiment. Under excitation, the response of the flow field had also made a certain change. Corresponding to the relative efficiency involved in the experiment, the separation position change was used to represent the positive and negative effects in numerical simulation. Compared with the description of efficiency gain, this method may be better in characterizing the changes of flow-field structure. Despite these inaccuracies, the curves calculated were overall a close match to the shape of the experimental data.

Corresponding to excitations with frequencies of three harmonics, the influences of unsteady excitations on vortex shedding flow were divided into three different types: Vortex-shedding control (SVS), suction-side separation vortex control (SSVS), and pressure-side separation vortex control (PSVS). The three types of separation flow control methods would be discussed separately in the following section.

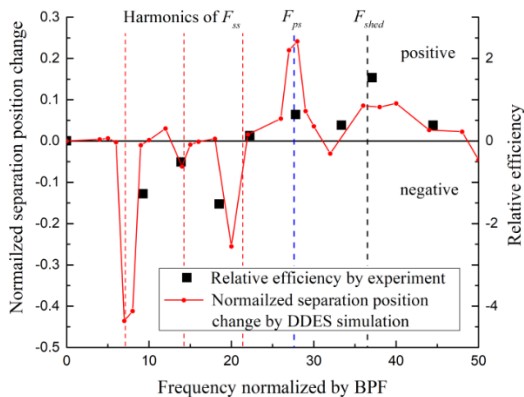

**Figure 13.** Effect of unsteady excitation at near stall point.

*4.4. Response of Separation Flow under Unsteady Excitations*

For the traditional method to control the separation of flow field, it was common to excite the separated flow with the frequency of shedding vortices. Under this condition the working range and total performance of the stage also had been improved, which had been widely investigated by researchers [9–13]. In this section, the excitation by the frequency of shedding vortices on the separation flow was also conducted. As Figure 13 indicated, positive effects were indeed obtained on the control of separation flow at a range of ±12.5% around $F_{shed}$.

As shown in Figure 14, the structure of the vortex under excitation was similar to the initial state, which was presented in Figure 6 without excitation. From the total pressure distribution in Figure 15, the separation zone and total pressure loss were all reduced with the unsteady excitation at the frequency of 36 times the RF. The mechanism of this phenomenon was that periodic excitation rectified the separation flow and injected energy into the vortex shedding flow. While increasing the strength of shedding vortices, the separation flow on the suction side of blade was suppressed. Then the pressure loss was also reduced. Inhabitations of the suction-side separation and rectification of the vortex shedding structure under SVC had been optimized. But along with the variation of flow rate, the characteristic frequency of wake shedding vortices also changed. It was very difficult to accurately excite the flow field with the frequency of wake shedding vortices at all operating points.

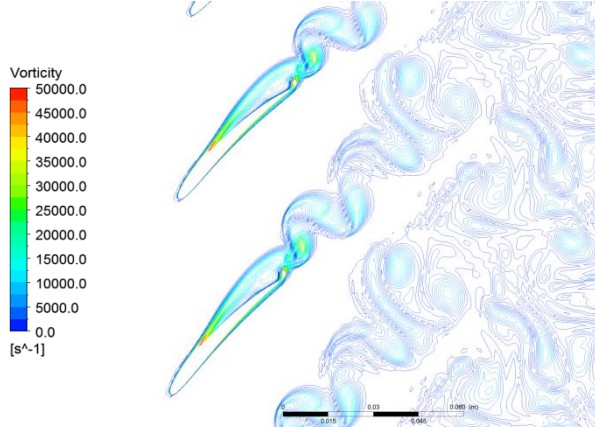

**Figure 14.** Instantaneous vorticity contour with excitation at 36 times the RF.

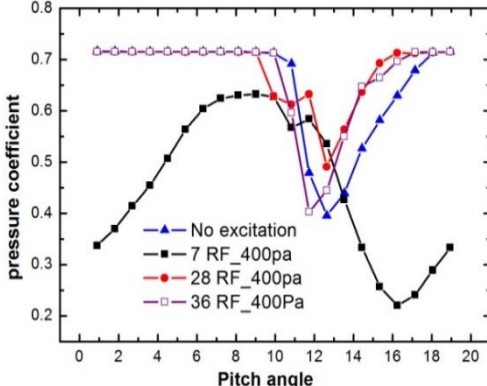

**Figure 15.** Pressure loss comparison with and without excitations.

For the suppression of suction-side separated flow, the effectiveness of the excitations with the frequency of suction-side separation vortices was checked as the next step. The vortex frequency of the suction surface was not the dominant frequency of the flow field, as indicated in Figure 7. The vorticity of the suction-side separation vortex was about an order magnitude lower than the vorticity of shedding vortices. However, after the excitation at ±5% of the range around the harmonics of $F_{ss}$, it was shown as a remarkable negative separation control effect, as expressed in Figure 13. In order to investigate the mechanism for the negative effect, the computation example with the excitation at the frequency of seven times of the RF was analyzed here. The instantaneous vorticity contour under excitation was presented in Figure 16. With the excitation, the vortex on the suction side separated forward to the leading edge of blade. And the structure of the vortices became disordered, and the flow field deteriorated markedly. The separation flow increased the flow instability and could even cause stall flow in the cascade. Loss distribution in Figure 15 showed that the pressure coefficient at the cascade exit dropped sharply, and the average pressure loss downstream of the blade increased significantly. Excitation that coincided with the eigenfrequency of suction-side separation transfused energy to the separation flow and thus led the separation position forward. This could cause the reduction of the working range and increase the loss in the flow field.

For the analysis above, it was concluded that the method with activating the flow field may not always have a positive effect. The effectiveness depended on the response of the separation flow field to excitations. Next, the excitations with the frequency of pressure side separation vortices (28 times the RF) were conducted to check whether it worked or not. From Figure 13, it could be seen that unsteady excitations at frequencies around the $F_{ps} \pm 7.5\%$ range had a magnificent effect on the separation control. Instantaneous separation vortex structure under PSVC (Figure 17) was very different from its inherent structure (Figure 6). The separation region of suction side vortices had been decreased,

and the strength of the shedding vortices also had been weakened. The comparisons of total pressure distribution with and without excitations in Figure 15 indicated that the core area of pressure loss moved to the pressure side of the blade, while the loss zone on the suction side decreases significantly with the unsteady excitation. With input of the excitation energy, the strength of the pressure-side separation vortices increased under the excitation. And because of the weak influence of the suction separation flow on the vortex shedding, the flow field deviated the vortex shedding to the pressure side and accelerated the shedding procedure under the increasing vorticity in pressure-side separation. As a result, the suction-side separation flow was restrained with a reduction of pressure loss. Even with the relative increase of pressure loss in the pressure side, the stage performance was still enhanced.

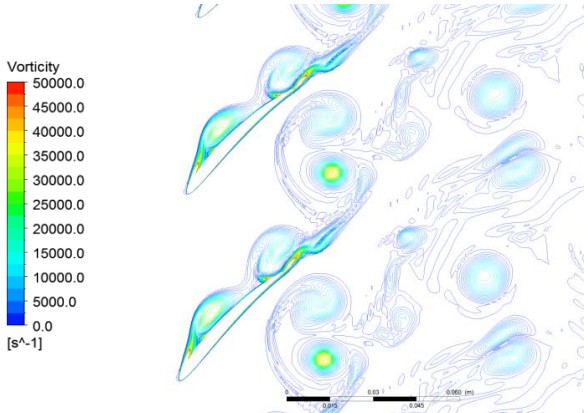

**Figure 16.** Instantaneous vorticity contour with excitation at 7 times the RF.

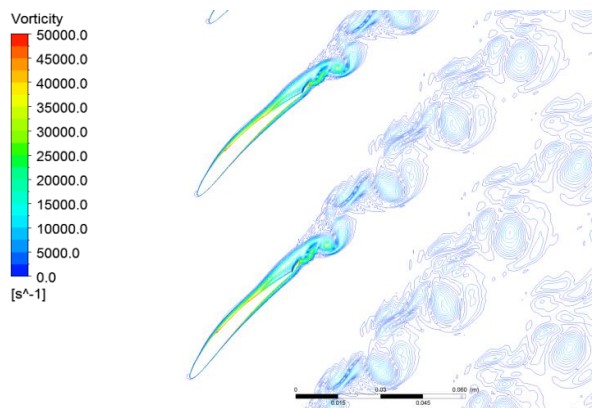

**Figure 17.** Instantaneous vorticity contour with excitation at 28 times the RF.

## 5. Discussion on the Application of PSVC

It could be pointed out that the maximum efficiency enhancement was obtained under the excitation with the frequency of shedding vortex from Figure 13. However, the frequency of unsteady excitation must be altered along with mass-flow coefficients to achieve the positive effects under vortex shedding control. As indicated in Figure 8, the vortex shedding frequency was sensitive to the mass-flow rate. The traditional separation control method, which was based on VSC, demanded the variation of frequency at excitation, altering with the working points. The effect of separation control was mainly dependent on the frequency of excitation. So, it was very difficult to apply in an industry application, which required alternating the frequency of wake shedding vortices needed at all operating points.

Meanwhile, the variations of the other two vortex separation flow frequencies $F_{ps}$ and $F_{ss}$ were relatively much smaller than the variation of the vortex shedding frequency to the change of working

conditions, as shown in Figure 6. Though the effective range of PSVC was less than that of VSC, it was highly possible to obtain a performance enhancement at the whole working range under a certain excitation frequency that equaled the effect under excitation of frequency $F_{ps}$.

To verify this conjecture, the computations with the excitation at the frequency of 28 times the RF were conducted with DES. And the effects on the stage performance under excitation were presented in Figure 18. It was shown that under the type of separation control of PSVC, separation positions on the suction side of blade were delayed almost in the whole working range. And the cascade performance was also enhanced. As a less investigated kind of separation control method, the PSVC was indeed able to obtain stage performance improvement at the whole working range with a certain frequency. This result turned out to be a case that it was easier to apply in industries by the PSVC than that of the traditional unsteady separation control method.

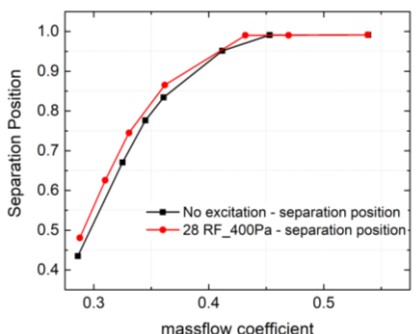 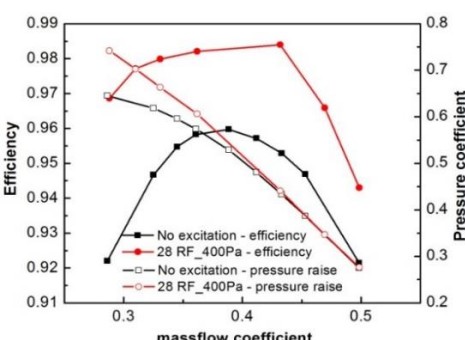

**Figure 18.** Variation of separation position on the suction side (**left**) and total performance (**right**) under pressure-side vortex control (PSVC).

## 6. Conclusions

In this paper, the control of the vortex shedding structure in a low-speed axial compressor model was analyzed using the DES method. Three different types of vortex control methods under unsteady excitations were classified by numerical results and previous experimental data.

The transformation of separation vortices in this test case indicated that, besides the traditional unsteady separation control method, excitation at other inherent separation frequencies could also have remarkable effects on the control of separation flow. The classifications of SSVC and PSVC effectively had been complementary to the separation control theory and demonstrated the potential capability on control of an unsteady vortex structure with neglected eigen frequencies. Specifically, unsteady excitations at frequencies around $F_{ps}$ had a significant separation control effect, while excitations at frequencies around $F_{ss}$ reduced the stability of separation flow.

The performance improvement obtained by the traditional unsteady control required the alteration of excitation frequencies at different working points, because the vortex shedding frequency was sensitive to the mass-flow rate. Meanwhile, the stage performance enhancement by PSVC, which had not been widely studied before, could be achieved at the whole working range by excitation at a certain frequency in the compressor test case.

**Author Contributions:** M.Z. conceived and designed the numerical simulation; M.Z. performed the simulation and analyzed the data; A.H. supervision and administration; M.Z. writing the original draft; A.H. reviewed and edited the final draft.

**Funding:** This research was funded by the National Science and Technology Major Project, grant number 2017-II-0009-0023.

**Conflicts of Interest:** The authors declare no conflict of interest.

## Abbreviations

The following abbreviations are used in this manuscript:

| $A_m$ | amplitude of unsteady excitation |
|---|---|
| $C_{DES}\Delta$ | local mesh spacing |
| $D_{DES}^k$ | dissipation rate in the equation for the turbulent kinetic energy in DES model |
| $F$ | frequency of unsteady excitation |
| $F_{ps}$ | frequency of pressure side vortex shedding |
| $F_{shed}$ | frequency of trailing edge vortex shedding |
| $F_{ss}$ | frequency of suction side vortex shedding |
| $P_{inlet}$ | total pressure at the stage inlet boundary |
| $l_{k-\omega}$ | local turbulent length scale in the SST $k - \omega$ turbulence model |
| $l_{DES}$ | local turbulent length scale in DES model |
| $k$ | turbulent kinetic energy |
| $t$ | physical time during the unsteady simulation |
| $\beta$ | SST Closure Constant |
| $\rho$ | density |
| $\Psi$ | mass-flow coefficient |
| $\omega$ | specific rate of dissipation |
| $\Delta p$ | pressure raise |
| $U_m$ | rotational speed at midspan |
| $\Delta C_p$ | pressure coefficient, $\Delta p / (\frac{1}{2}\rho U_m^2)$ |
| $\Delta$ | averaged wall shear stress |
| $C_\delta$ | shear stress coefficient, $\delta / (\frac{1}{2}\rho U_m^2)$ |

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
