# Peer review of "Numerical Investigation on Unsteady Separation Flow Control in an Axial Compressor Using Detached-Eddy Simulation"

_applsci, doi:10.3390/app9163298_

Round 1
Reviewer 1 Report
The paper provides interesting information on compressure stage flow control diminishing trailing edge flow separation by frequency dependent forcing. The analysis is perfomed employing DES with respect to experiments conducted on a compressure stage test rig.
The paper is well written including the relevant data.
A minor revision is recommended mainly attributed to the fact that the quantities discussed are presented in dimensional form. Typically, such data are given in non-dimensional form for general analysis and comparison with literature data. Consequently, the pressure raise (Fig. 3), wall shear stress (Fig. 7) and total pressure (Figs. 9, 13, 15, 17) should be given as coefficients including the formulas for the non-dimensonal form. Also, the vorticity (Figs. 4, 5, 12, 14, 16) should be given in non-dimensional form and it should be clarified that the vorticity magnitude is shown.
Some final proof reading with respect to Englisch wording is also recommended
(e.g. vortexes -> vortices, etc.)
Author Response
Reply to Reviewer
Paper: applsci-552960
Title: Numerical Investigation on unsteady separation flow control in an axial compressor using Detached-Eddy Simulation
Thank you very much for the suggestions for our draft. We have replied point by point to your comment, along with a clear indication of red color at the location of the revised paper.
The comments and replies can be summarized as follows:
For the quantities mentioned in your suggestion, such data have been given in non-dimensional form. The pressure raise and total pressure have been changed to pressure coefficient. The mass-flow coefficient is defined as the ratio of averaged axial velocity to averaged total velocity. The wall shear stress is also changed to shear stress coefficient. The formulas for the non-dimensonal form of corresponding parameters are defined in the revision.
In continuum mechanics, the vorticity is a pseudovector field defined as the curl of the flow velocity vector. It describes the local spinning motion of a continuum near some point. The definition is also expressed by the vector analysis formula in the revision.
The typographical and grammatical errors had been revised most in the revised paper. We have made the correction accordingly.
Thanks for your advice.

Reviewer 2 Report
Dear Authors,
The authors investigate the response of a blade cascade flow to unsteady forcing prescribed by a sinusoidal fluctuation at a given frequency of the total pressure. They use numerical analysis and they focus on the efficiency and the separation position of the flow at three different positions.
The paper is pitched as a flow control paper, but it is not the case. It is a study on the dynamical response of the flow to unsteady forcing. As vortex shedding is similar to an autonomous oscillator, the flow will response mainly to its eigenfrequency with an amplification factor, as you have shown.
The most interesting part is section 5.4 and I suggest to focus the paper on this section with a better control strategy.
There is clearly a lack of originality in this paper. Therefore, I don't recommend this manuscript to be published.
Author Response
Reply to Reviewer
Paper: applsci-552960
Title: Numerical Investigation on unsteady separation flow control in an axial compressor using Detached-Eddy Simulation
Thank you very much for the suggestions for our draft. We have replied point by point to your comment, along with a clear indication of red color at the location of the revised paper.
The comments and replies can be summarized as follows:
A lot of efforts had been studied to the separation control with vortex shedding. But little attention has been paid to the control of flow field separation by controlling vortices of blade pressure side. It could be concluded that the vortex shedding frequency Fshed varied more than 100% with mass-flow coefficients, while the frequencies Fps was insensitive to the variation of working conditions. Traditional separation control method which was based on SVC demanded the variation of frequency at excitation altering with the working points. So it was very difficult to apply in industry application which required alternating the frequency of wake shedding vortices needed at all operating points. In this paper, it aimed to investigate the effect on the separation control by the excitation with the frequency of Fps. It was shown that under the type of separation control of PSVC, separation positions on the suction side of blade were delayed almost in the whole working range. And the cascade performance was also enhanced. As a less investigated kind of separation control method, the PSVC was indeed able to obtain stage performance improvement at the whole working range with a certain frequency.
As for your suggestion, the overall structure of the paper was changed. The part which was in the section 5.4 of manuscript was adjusted to the application of pressure side vortex control in order to highlight its importance.
Reviewer 3 Report
Please see the attached review pdf.
Review of the Manuscript ID: applsci-552960, titled
“Numerical investigation on unsteady separation flow control in an axial compressor using
Detached-Eddy Simulation”
– byMZhang and A Hou
(under consideration for publication at the Applied Sciences)
This manuscript presents numerical results on vortex shedding in compressors, under unsteady
excitation at different frequencies. The findings reveal new information on pressure side vortex
shedding and its correlation with the background flow. While this is an important study, I
recommend that the authors consider the following suggestions/edits which might improve the
manuscript.
1. The numerical simulations use DES (Detached-Eddy SImulation) scheme for the turbulence
modeling. However, it is well-known that the averaging implemented by the k-! set-up does
a poor job of capturing short time-scale effects, such as vortices, that frequently emerge particularly
when the flow rates are high. These vortices have a significant effect on the overall flow
physics. So, a major current push in the field has been to upgrade the simulations to (at least)
LES (Large Eddy Simulations), for a more accurate description (particularly when the turbulent
scales are dominant). Thus, I recommend that the authors use LES for more robust results.
May be just a trial comparison between LES and DES results would suffice, with a good agreement
therein justifying the continued use of DES as the more computationally inexpensive yet
accurate-enough option.
2. I do not understand equation 4. The different terms in the equation are dimensionally different.
3. I would suggest adding more information on all the figure captions.
4. Finally, I would advise a thorough re-check for typos and grammatical errors. There are many
grammatical errors and sentence construction flaws, that should be easy enough to remove, and
would enhance the manuscript’s readability.
In conclusion: I reiterate that this work can be an important contribution. But, before I can recommend
its publication, I would like the authors to consider my suggested edits.

Author Response
Reply to Reviewer
Paper: applsci-552960
Title: Numerical Investigation on unsteady separation flow control in an axial compressor using Detached-Eddy Simulation
Thank you very much for the suggestions for our draft. We have replied point by point to your comment, along with a clear indication of red color at the location of the revised paper.
The comments and replies can be summarized as follows:
Indeed Large Eddy Simulation is accurate for capturing vortices, but it needs much more computation cost. Detached-Eddy Simulation (DES) is a hybrid large eddy simulation (LES)/RANS method. Compared with using LES in the global computational domain, the DES method not only ensures the calculation accuracy, but also improves the calculation efficiency and saves the calculation cost. And the DES method had been widely applied in capturing vortex-shedding structures in turbomachines. Due to the limitation of computing time, the DES was used in this research. But LES is in consideration for the following research.
To imitate the inlet unsteady excitation, the pressure varying with time was set up at inlet. The formula was defined as equation 4. Its definition can be set to the superposition of a steady value and an unsteady value varying with time. The amplitude Am was the maximum excitation amplitude of pressure at the inlet. And f was the frequency of the excitation.
In order to analyze the impact of excitation on vortex shedding structure, much more analysis had been performed under unsteady excitations at different frequencies on all the figure captions in the compressor model.
The typographical and grammatical errors had been revised most in the revised paper. We have made the correction accordingly.
Thanks for your advice.
Round 2
Reviewer 2 Report
Dear authors,
The quality of the manuscript has been significantly improved with the new modifications. However, I have still a few questions that need relevant answers.
Line 140, it is written that the maximum excitation amplitude is 600 Pa. However, in figure 12 you performed DES at 1000 Pa. Can the authors clarify this? Line 143, the computations are performed for a near-stall operating condition. The operating condition must be defined with numbers. Line 182, DES simulations have been performed at the working conditions A to investigate the flow separation. However, figure 5 shows the results for different mass flows. Where these results come from? If it is DES, it should be written somewhere that different operating conditions have been investigated and if the mesh used can properly reproduce experimental data. The units of the plotted quantities must be specified in all figures. In figure 5, the y-axis label must be corrected. In figure 7, the color of the boxes is not consistent. In legend 10, it will be better to define the separation position inside the legend also. In legend 11, it will be better to define psi inside the legend. Can the authors discuss the discrepancy between the experimental result and the simulation at F_ps in figure 13? In the discussion, the authors claim that it is easier to apply a constant excitation than a tuned frequency following the vortex shedding frequency. But how it can be practically implemented? Why the curves in figure 5 and 18 are not the same? In figure 18, I remark that the maximum efficiencies between this version and the previous version of the manuscript are not the same. Is this correct? In figure 18, is the mesh used in the DES relevant to all operating conditions? Because the validation has been performed for a specific operating condition.
As general remarks, the quality of the figures must be improved and there are English typos that need to be corrected.
Author Response
Reply to Reviewer
Paper: applsci-552960
Title: Numerical Investigation on unsteady separation flow control in an axial compressor using Detached-Eddy Simulation
Thank you very much for the suggestions for our draft. We have replied point by point to the comments, along with a clear indication of red color at the location of the revised paper.
The comments and replies can be summarized as follows:
Q: Line 140, it is written that the maximum excitation amplitude is 600 Pa. However, in figure 12 you performed DES at 1000 Pa. Can the authors clarify this?
A: In line 140(maybe line 126), according to the experiment the maximum excitation amplitude was set to be 600 pa. But in the computations, the maximum amplitude was extended to 1000 pa to investigate the influence of excitation amplitude on efficiency and separation position. To avoid misunderstanding, these descriptions were made corrections. (line127-129)
Q: Line 143, the computations are performed for a near-stall operating condition. The operating condition must be defined with numbers.
A: The computation boundary conditions were set up based on the parameters of a near-stall working state with a total pressure rise over 1600 Pa in the calculation. (line131)
Q: Line 182, DES simulations have been performed at the working conditions A to investigate the flow separation. However, figure 5 shows the results for different mass flows. Where these results come from? If it is DES, it should be written somewhere that different operating conditions have been investigated and if the mesh used can properly reproduce experimental data.
A: The DES simulations have been performed at different mass-flow coefficients. The result at the working conditions A was selected to investigate unsteady separation flow structure. These descriptions were added in the revision. (line158, 169-171)
Q: The units of the plotted quantities must be specified in all figures. In figure 5, the y-axis label must be corrected. In figure 7, the color of the boxes is not consistent.
A: The data in figures were given in non-dimensional form for general analysis. So the y-axis label of pressure raise was set as pressure coefficient. And the color of the boxes in figure 7 was adjusted in the revision.
Q: In legend 10, it will be better to define the separation position inside the legend also. In legend 11, it will be better to define psi inside the legend.
A: The separation position on the suction side was identified where the wall shear stress on the suction side switched from positive to negative. In figure 10, the y-axis label of separation position was the chordwise of the blade. Indeed the unit of psi was used to express the pressure somewhere. For consistency in this paper the pressure coefficient was chosen to indicate the distribution of pressure at exit. Thanks for your advice. (line215-217)
Q: Can the authors discuss the discrepancy between the experimental result and the simulation at F_ps in figure 13?
A: For the difference of frequencies and effects between the experimental and computed results, there were two possible causes. Because of the limitation of CFD in simulating the turbulent flow, the predicted structure of vortices still existed a little discrepancy with the experiment. Under the excitation the response of flow field had also made a certain change. Corresponding to the relative efficiency involved in the experiment, the separation position change was used to represent the positive and negative effects in numerical simulation. Compared with the description of efficiency gain, this method may be better in characterizing the changes of flow field structure. Despite these inaccuracies, the curves calculated were overall a close match in the shape to the experimental data. (line260-268)
Q: In the discussion, the authors claim that it is easier to apply a constant excitation than a tuned frequency following the vortex shedding frequency. But how it can be practically implemented?
A: In this paper, it mainly aimed to investigate the effect on the separation control by the excitation with the frequency of Fps, and explore the mechanism of that. For the methods and implements of excitation, they can be unsteady suction blowing, upstream wake excitation, trumpets and so on. In the experiment, the periodic sound excitation was generated by a pneumatic trumpet. And in the following research we will consider the implement to take in practice.
Q: Why the curves in figure 5 and 18 are not the same? In figure 18, I remark that the maximum efficiencies between this version and the previous version of the manuscript are not the same. Is this correct? In figure 18, is the mesh used in the DES relevant to all operating conditions? Because the validation has been performed for a specific operating condition.
A: In figure 18, the calculation had been extended to a larger flow rate. So it looked like different to figure 5. For a better comparison, the coordinate values of the y-axis had been changed accurately to the same. According to the comment of another reviewer, the pressure value of the y-axis in the manuscript had been change to the pressure coefficient in the revision. Except for this, the rest of figure 18 remained the same. And the computations with the excitation shown in figure 18 were all conducted with DES. This had been mentioned in the new revision. (line350-351, figure 5, 18)
The typographical and grammatical errors had been revised most in the revised paper. We have made the correction accordingly.
Thanks for your advice.

Reviewer 3 Report
Thanks for the responses.
Author Response
Thank you very much for the excellent work and suggestions on our manuscript.